# Diffusion-Tensor-Tractography-Based Diagnosis for Injury of Corticospinal Tract in a Patient with Hemiplegia Following Traumatic Brain Injury

**DOI:** 10.3390/diagnostics10030156

**Published:** 2020-03-13

**Authors:** Chan-Hyuk Park, Su-Hong Kim, Han-Young Jung

**Affiliations:** Department of Physical &amp, Rehabilitation Medicine, Inha University School of medicine, Inha University Hospital, Incheon 22332, Korea; chanhyuk@gmail.com (C.-H.P.); suhong1207@gmail.com (S.-H.K.)

**Keywords:** traumatic brain injury, diffusion tensor imaging, traumatic axonal injury, corticospinal tract, hemiplegia

## Abstract

This paper reports a mechanism for corticospinal tract injury in a patient with hemiplegia following traumatic brain injury (TBI) based on diffusion tensor tractography (DTT) finding. A 73-year-old male with TBI resulting from a fall, without medical history, was diagnosed as having left convexity epidural hematoma (EDH). He underwent craniotomy and suffered motor weakness on the right side of the body. Two weeks after onset, he was transferred to a rehabilitation department with an alerted level of consciousness. Four weeks after onset, his motor functions were grade 1 by the Medical Research Council’s (MRC) standards in the right-side limbs and grade 4 in the left-side limbs. The result of DTT using the different regions of interest (ROIs) showed that most of the right corticospinal tract (CST) did not reach the cerebral cortex around where the EDH was located, and when the ROI was placed on upper pons, a disconnection of the CST was shown and a connection of the CST in ROI with the middle pons appeared. However, the right CST was connected to the cerebral cortex below the pons regardless of ROI. This study is the first report to use DTT to detect that the discontinuation of the left CST in the cerebral cortex and injury lesions below the lower pons and between the upper and lower pons are responsible for motor weakness in a patient.

## 1. Introduction

Neural injury in white matter is responsible for neurological disorders following traumatic brain injury (TBI) [1,2]. Many previous studies reported injury of the corticospinal tract (CST), which plays an important role in motor control, as is common in patients with TBI [1,2,3,4]. The prevalence of motor weakness after head injury was reported to range from 9% to 56% [5].

Diffusion tensor tractography (DTT) can provide information about neural tracts not available by conventional magnetic resonance imaging (MRI) [1,6,7]. In particular, DTT allows access to injury mechanisms of traumatic axonal injury (TAI) following TBI, useful for detecting axonal injury lesions that are not detected by conventional brain MRI (T1-weighted, T2-weighted, fluid attenuated inversion recovery (FLAIR)), and GRE (T2-weighted gradient recall echo) images [1,2,5,8]. Thus, the aim of this report is to outline a mechanism that can detect lesions of CST injuries in a hemiplegic patient following TBI based on DTT findings with different regions of interest (ROIs).

## 2. Case Report

A 73-year old right-handed man without a history of sequelae three years before his TBI, and without a relevant medical history was the subject of this study. He recently visited Inha University Hospital’s emergency room (Incheon, Korea) due to a drowsy mental status immediately after hitting the temple of his head on the ground during a fall while inebriated. On admission, he suffered from right-side weakness and showed no loss of consciousness (Glasgow coma scale (GCS): 9/15). Initial brain computed tomography (CT) showed a large acute epidural hematoma (EDH) of 4.1 × 10.3 cm at the left convexity and mild subfalcine herniation in the right region with chronic contusion in the right frontal and left anterior frontal lobes (Figure 1A). The hematoma was removed by emergency craniotomy in our neurosurgery department. After surgery, he persistently complained of right-side weakness, and two weeks after onset, he was transferred to our rehabilitation department. Upon transfer, he showed an alert level of consciousness (GCS: 14/15), no post-traumatic amnesia, and complained of severe weakness in the right upper and lower extremities (Medical Research Council (MRC) scale: upper extremities 1/5, lower extremities 1/5). A brain MRI was conducted four weeks after onset, which showed increased subdural hematoma in the left frontal and parietal lobes with encephalomalacia due to previous contusion hemorrhage in both the frontal lobes and EDH resolution. In addition, multiple petechial hemorrhages were observed in right cerebellum, pons, both thalami, both basal ganglia, and left temporal lobe, and these evidenced diffuse axonal injury, probably due to hypertensive microbleeding (Figure 1B,C). He took medications including memantine for cognitive impairment and carvedilol for blood pressure control, and received conventional rehabilitation. This patient provided informed consent. 

## 3. Diffusion Tensor Tractography

DTT images were acquired using a 3.0 T GE Signa Architect MRI System (General Electric, Milwaukee, WI, USA) using the following conditions: slice thickness = 2 mm, 30 directions, and 72 contiguous slices field of view = 240 × 240 mm, acquisition matrix of 128 × 128, b = 1000 mm^2^s^−1^, repetition time (TR) = 15,000 ms, and echo time (TE) = 80.4 ms. DTI studio software (Johns Hopkins Medical Institute, Baltimore, MD, USA) was used to analyze DTT images. We reconstructed two CSTs. Tract 1 conditions were ‘OR’ ROI on the lower anterior pons and ‘AND’ ROI on middle anterior pons. ROIs for Tract 2 Target ROI were placed on anterior upper pons for ‘AND’ ROI and lower anterior pons for ‘OR’ ROI. Fiber tracking was conducted at a fractional anisotropy (FA) of <0.2 and a turning angle of >60°. Mean diffusivities (MDs), tract volume (TVs), and FAs were obtained from DTT images of the CST. We compared the subject’s CSTs with those of six age-matched healthy control subjects (2 men and 5 women with mean age 71.83 years, range 68 to 78 years). A statistical analysis for the determination of differences in FA, MD, and TV of two tracts in control subjects was performed in two independent sample t-test using SPSS software (v. 22.0; SPSS, Chicago, IL, USA). The results showed no significant difference between FA, MD, and TV values of the two tracts in control subjects (*p* > 0.05). Four weeks after onset, the left CST of Tract 1 was noticeably thinner than the right CST and discontinuous with the cerebral cortex. (Figure 2A,Cb). In Tract 2, the left CST showed a discontinuation to the cerebral cortex and a disconnection below the lower pons compared to Tract 1 (Figure 2B,Ca). As in previous studies, DTT parameter values (FA, MD, and TV) indicating a deviation of more or less than two standard deviations(SDs) compared with control subjects were defined as abnormal [9,10]. The majority of the left CST exhibited lower FA and TV values than SDs and higher MD values than two SDs. However, those of the right CST were not significantly different in Tract 1 and Tract 2 compared with control subjects (Table 1). 

## 4. Discussion

Our findings demonstrated neural injury (discontinuation, narrowing, and disconnection) of the CST using DTT images in a patient that experienced motor weakness of right extremities after TBI. The CST plays an important role in motor control in the extremities and is the representative descending tract that passes through the corona radiate, the posterior limb of the internal capsule, and the cerebral peduncles. This is decussated into the lateral spinal cord. The damage of the CST results in motor weakness [1,3,11]. In this case, conventional MRI (GRE) did not detect traumatic axonal injury (TAI) where the CSTs passed. In contrast, in the DTT result using two methods, our patient showed that when that target ROI was placed on the middle pons, the left CST was thinner than the right CST and exhibited discontinuation to the cerebral cortex around the EDH location (Tract 1), but the tract of the left CST using the target ROI on the upper pons was disconnected below the lower pons with a discontinuation at the cerebral cortex (Tract 2). However, the right CST did not differ between the two tracts. Therefore, in this study, although conventional MRI in our patient revealed no specific lesion where the left CST passed, using DTT, we identified two injury lesions where the left CST passed: one lesion below the lower pons and another lesion between the upper pons and middle pons (Figure 2Cc).

The left CST had a higher MD value but lower FA and TV values than control subjects, whereas the right CST did not show significant differences compared with control subjects. The FA values reflect microstructural directions and integrities of axons, myelin, and microtubules [9,12,13,14,15]. MD provides a quantitative measure of water motion, which reflects pathological changes in white matter [16]. Increases in MD values represent vasogenic edema or axonal injury [2,16]. TVs indicate information about numbers of voxels contained within neural tracts [9]. In other words, low FA and TV values and high MD values are indicative of neural injury, and reflect the presence of diffused axonal injury [1,2,8]. Therefore, in our patient, low FA and TV values and high MD represented neural tract injury of the left CST as TAI. 

In conclusion, to the best of our knowledge, this is the first study conducted with DTT using different target ROIs to address injury lesions not observed in conventional MRI in a patient with right hemiplegia following TBI. The discontinuation of CST in the left cerebral cortex and two injury lesions on the left pons were responsible for motor weakness. As in previous studies, our findings also suggest that the analysis of CSTs is useful to observe lesions of TAI after TBI [5]. However, the present study has several limitations including: (1) the case report design, so larger-scale studies are required; (2) long-term follow up would be required to determine motor recovery; and (3) DTT depends on operator handling. [17]. In contrast, this DTT-based study reports significant injury lesions of the neural tract in a patient with motor weakness following TBI.

## Figures and Tables

**Figure 1 diagnostics-10-00156-f001:**
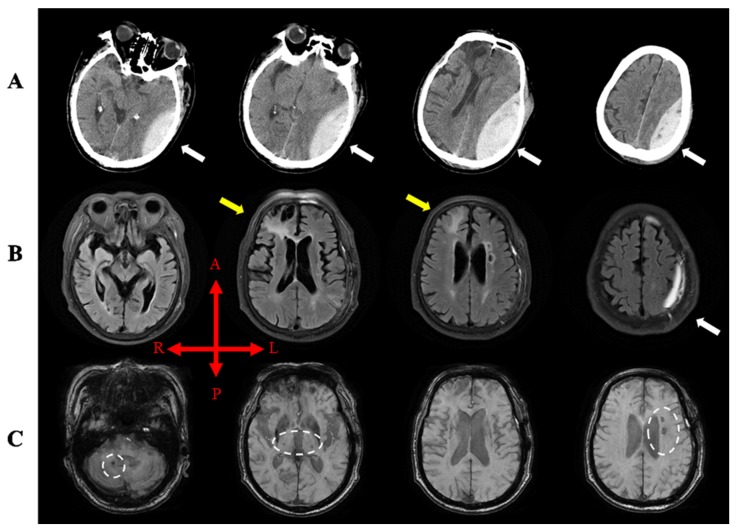
(**A**) Computed tomography (CT) images showing EDH caused by TBI (white arrow). (**B**) Magnetic resonance T2 FLAIR image four weeks after onset showed encephalomalacia due to contusion hemorrhage in the right frontal lobe (yellow arrow) and increased SDH in the left frontal and parietal lobes (white arrows). (**C**) Multiple petechial hemorrhages with diffused axonal injury on GRE images were observed in right cerebellum, pons, both thalami, both basal ganglia, and left temporal lobe (white dashed circle). Note: EDH, epidural hematoma; TBI, traumatic brain injury; FLAIR, fluid attenuated inversion recovery; GRE, T2-weighted gradient recall echo; R, right; L, left; A, anterior; P, posterior.

**Figure 2 diagnostics-10-00156-f002:**
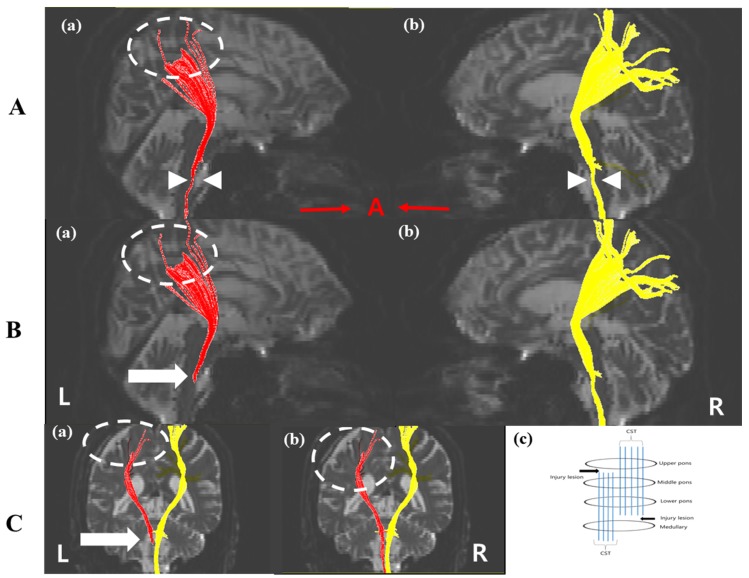
Diffusion Tensor Tractography images four weeks after onset. (**A**) Tract 1 which used ROIs on the lower pons and middle pons showed that (a) the left CST was thinner than the right CST (white arrow head) and (b) discontinuation in the cerebral cortex in the left CST appeared (white dashed circle). (**B**) Tract 2 using ROIs on the lower pons and upper pons indicated the thinner left CST (a) compared with than the right CST (b) did not reach the cerebral cortex (white dashed circle) and the disconnection below the lower pons is shown (white arrow). (**C**) The disruption of the left CST around the EDH location (dashed circle) (a,b) and the dissociation of the left CST below the lower pons (white arrow). (c) Injury lesions analyzed by DTT results. Injury lesions were below the lower pons and between upper and lower pons. CST: corticospinal tract, EDH: epidural hematoma, ROI: region of interest, DTT: diffusion tensor tractography.

**Table 1 diagnostics-10-00156-t001:** DTT parameter of the corticospinal tracts of the patient and control subjects.

		FA	TV	MD (×10^−3^ mm^2^/s)
Patient	Tract 1			
	Right	0.6015	3892	0.771
	Left	0.5029 **	1396.000 **	0.839 **
	Tract 2			
	Right	0.6014	4071.000	0.770
	Left	0.4997 **	1449.000 **	0.840 **
Controls (*n* = 7)				
Tract 1	Subject 1	0.621	5017.500	0.732
	Subject 2	0.596	4992.000	0.741
	Subject 3	0.661	4874.000	0.679
	Subject 4	0.637	5359.500	0.727
	Subject 5	0.610	5180.500	0.736
	Subject 6	0.654	5321.500	0.669
	Subject 7	0.654	5430.000	0.689
	Mean (SD)	0.633 (0.027)	5167.857 (447.741)	0.710 (0.032)
Tract 2	Subject 1	0.617	5457.000	0.727
	Subject 2	0.597	4924.500	0.741
	Subject 3	0.659	4451.500	0.676
	Subject 4	0.635	4344.000	0.725
	Subject 5	0.609	5518.000	0.740
	Subject 6	0.653	5584.000	0.673
	Subject 7	0.653	5338.500	0.683
	Mean (SD)	0.632 (0.026)	5088.214 (616.730)	0.710 (0.034)

CST, corticospinal tract; FA, fractional anisotropy; TV, tract volume; MD, mean diffusivity; SD, standard deviation. Tract 1: ‘OR’ ROI on lower pons and ‘AND’ ROI on middle pons, Tract 2: ‘OR’ region of interest (ROI) on lower pons and ‘AND’ ROI on upper pons. ** Parameters two SDs above or below that of control subjects.

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
