# Peer review of "Diffusion-Tensor-Tractography-Based Diagnosis for Injury of Corticospinal Tract in a Patient with Hemiplegia Following Traumatic Brain Injury"

_diagnostics, 2020, doi:10.3390/diagnostics10030156_

Round 1
Reviewer 1 Report
This case report showed neural injury (discontinuation, narrowing, and disconnection) of the CST using DTT images in a patient that experienced motor weakness of right extremities after TBI. Overall, the dscription of the case is well and the manuscript is well description. I just have a minor suggesetion.
It is better to send the manuscript for English edition by a native speaker.
Author Response
This case report showed neural injury (discontinuation, narrowing, and disconnection) of the CST using DTT images in a patient that experienced motor weakness of right extremities after TBI. Overall, the dscription of the case is well and the manuscript is well description. I just have a minor suggesetion.
It is better to send the manuscript for English edition by a native speaker.
Ans> This manuscript edited English by native speaker in MDPI
Reviewer 2 Report
Park et al. have written an interesting case report from the human subject entitled, " Diffusion tensor tractography based diagnosis for injury of corticospinal tract in a patient with hemiplegia following traumatic brain injury". The strength of the article is that- 1) The author assessed the mechanism for corticospinal tract injury in the patient with hemiplegia after TBI exploiting DTI methods, and they determined that there is discontinuation of the left CST at cerebral cortex and injury lesions around pons that could result in the motor weakness
in the patient
2)The article assessed the motor function of the patient and graded it.
3)There are few weakness in the article which are as below. a) Line 12-13: Please also mention here the patient history or complications if any. b) Was any drugs given for the patient during treatments including anesthetics, please explain in details? c) Line 34, 35: There are many parenthesis and some of those brackets are not properly close. I suggest to write long form outside and short form inside parenthesis. Line 40: Please replace "he visited our emergency room" with further information. Example: Was he admitted instantly after injury? What is the name and location of hospital? Was consent taken from the patient/institutional approval followed to publish this article? please mention as well. Line 43: Based on the patients symptoms, you first performed CT scan and transitioning sentences is not great. You should make the logical flow of the text here. Line 68: Can you provide the data for healthy control subjects (2 men and 5 women)? Line 73: Which test was used for calculation of statistical significance? please mention with all those p-values.
Line 124-125: As you mentioned that it is the first study of this kind, you should also highlight those points in your abstract.
Author Response
Please, "bold: required questions and Ans>: answer" checked.
a) Line 12-13: Please also mention here the patient history or complications if any.
Ans>
We revised the sentence as follows; A 73-year-old male with TBI resulting from falling down and without a medical history was diagnosed as havingto have left convexity epidural hematoma (EDH).
b) Was any drugs given for the patient during treatments including anesthetics, please explain in details?
Ans>
We add the medication at Line 107-108
He took medications including memantine for cognitive impairment and carvedilol for blood pressure control, and received conventional rehabilitation. This patient provided informed consent
c) Line 34, 35: There are many parenthesis and some of those brackets are not properly close. I suggest to write long form outside and short form inside parenthesis.
Ans> we revised Line 34, 35. Please check the sentence.
Line 40: Please replace "he visited our emergency room" with further information. Example: Was he admitted instantly after injury? What is the name and location of hospital? Was consent taken from the patient/institutional approval followed to publish this article? please mention as well.
Ans>
We revised line 40 as Recently, he visited our Inha University Hospital emergency room (Incheon, Korea) at drowsy mental status immediately after hitting his temporal head on the ground during a fall while inebriated (Glasgow coma scale (GCS): 9/15).
- Name, country, city, and immediately
Line 43: Based on the patients symptoms, you first performed CT scan and transitioning sentences is not great. You should make the logical flow of the text here.
Ans> we revised the sentence. Firstly Physical status was suggested, and after that, we suggested CT scan.
On admission, he suffered from right side weakness and showed no loss of consciousness (Glasgow coma scale (GCS): 9/15). Initial Bbrain computed tomography (CT) showed a large acute epidural hematoma (EDH) of 4.1 x× 10.3cm at the left convexity and mild subfalcine herniation to the right region with chronic contusion at the right frontal and left anterior frontal lobes (Figure 1A).
Line 68: Can you provide the data for healthy control subjects (2 men and 5 women)?
Ans> Please check the table 1
Line 73: Which test was used for calculation of statistical significance? please mention with all those p-values.
Ans> Line 73:Statistical significance was suggested as follows; A statistical analysis for the determination of differences in FA, MD, and TV of two tracts in control subjects was performed in two independent sample t-test using SPSS software (v. 22.0; SPSS, Chicago, IL, USA). The results showed no significant difference between FA, MD, and TV values of the two tracts in control subjects (p > 0.05).
Line 124-125: As you mentioned that it is the first study of this kind, you should also highlight those points in your abstract.
Ans> we revised abstract
“This study is the first report that using DTT, the discontinuation of the left CST at cerebral cortex, and injury lesions below lower pons and between upper and lower pons being responsible for motor weakness in a patient.” was added.